# miR-9 and miR-181a Target Gab2 to Inhibit the Proliferation and Migration of Hepatocellular Carcinoma HepG2 Cells

**DOI:** 10.3390/genes13112152

**Published:** 2022-11-18

**Authors:** Lantang Huang, Ruimin Liu, Peiyi Zhou, Yingpu Tian, Zhongxian Lu

**Affiliations:** 1Xiamen City Key Laboratory of Metabolism, School of Pharmaceutical Sciences, Xiamen University, Xiamen 361005, China; 2Fujian Provincial Key Laboratory of Innovative Drug Target Research, School of Pharmaceutical Sciences, Xiamen 361005, China; 3School of Pharmaceutical Sciences, Xiamen University, Xiamen 361102, China

**Keywords:** hepatocellular carcinoma, miR-9, miR-181a, Gab2

## Abstract

The incidence of liver cancer ranks seventh globally, with nearly half of all cases occurring in East Asia, but currently, there are very few drugs to treat it. Our previous studies demonstrated that the signal integration protein Gab2 is a potential drug target for the prevention and therapy of liver cancer. Here, we screened for and identified two miRNAs that target Gab2 to suppress the proliferation and migration of hepatocellular carcinoma (HCC) cells. First, we predicted Gab2-targeting miRNAs through biological websites, and we selected nine miRNAs that were reported in the literature as being abnormally expressed in liver cancer and fatty liver tissue. Then, we measured the expression of these miRNAs in the hepatic epithelial cell line HL-7702 and the HCC cell line HepG2. The expression levels of miR-9, miR-181a, miR-181c, miR-34a, and miR-134 were high in HL-7702 cells but low in HepG2 cells, and their expression patterns were the opposite of Gab2 in these cells. Furthermore, we transfected miR-9, miR-34a, miR-181a, and miR-181c mimics into HepG2 cells and found that only miR-9 and miR-181a reduced the level of Gab2 proteins. miR-9 also reduced the Gab2 mRNA level, but miR-181a did not affect the Gab2 mRNA levels. Using a miRNA-Gab2 3′UTR binding reporter, we confirmed that miR-9 and miR-181a bind to the Gab2 3′UTR region. Finally, we introduced miR-9 and miR-181a mimics into HepG2 cells and found that cell proliferation and migration were significantly inhibited. In conclusion, we identified two novel miRNAs targeting Gab2 and provided potential drug targets for the prevention and treatment of liver cancer.

## 1. Introduction

Liver cancer is the seventh most common cancer worldwide, with the highest incidence in east Asia [1,2]. It also has a very high mortality rate and is the second leading cause of cancer deaths [1,2]. Hepatocellular carcinoma (HCC) and cholangiocarcinoma are the main subtypes of liver cancer and are often caused by infection with hepatitis B and C viruses (HBV and HCV), persistent aflatoxin exposure, alcohol consumption, smoking, metabolic abnormalities, and, to a lesser extent, genetic mutations [1,3,4]. Hepatocellular carcinoma (HCC) mainly develops from chronic diseases, including alcoholic and nonalcoholic steatohepatitis, diabetes mellitus, metabolic syndrome, and viral hepatitis [5]. These chronic liver diseases affect a large number of patients, resulting in the frequent occurrence of liver cancer; liver disease has become a major disease endangering human health [1,4,5]. The treatment methods for HCC mainly include tumor resection, local therapy for non-resectable tumors, and liver transplantation [4]. However, these treatments are not effective because most HCC patients are still in an advanced stage of disease [6,7]. Molecular targeted therapy and the recently emerging immunotherapy approaches have good prospects, but there are still few clinical applications, with only the drug sorafenib currently available [3,4,7]. As a result, the five-year survival rate of HCC patients has not risen in the past several decades [1,4,7]. Therefore, new preventive and therapeutic targets and methods continue to be sought.

A signal scaffold protein, Gab2, was demonstrated to be a potential preventive and therapeutic target for liver diseases in recent studies [8,9,10,11]. Gab2 amplifies and integrates signal transduction to regulate cell growth, migration, differentiation, and apoptosis [12]. The Gab2 protein exhibits low expression in normal tissues and is involved in few physiological functions, such as bone development [10]. However, Gab2 was found to be recruited by a variety of pathogenic factors to initiate the pathogenesis of hepatitis, fatty liver, and obesity by mediating a variety of signaling molecules, thus promoting the occurrence and development of liver cancer [9,10,11]. Deletion of the *Gab2* gene and the suppression of Gab2 expression ameliorates the pathogenesis of hepatitis, fatty liver, and obesity and suppresses the development of liver cancer [9,10,11]. Gab2 is active in a variety of cancers, such as breast cancer [13], ovarian cancer [14], lung cancer [15], leukemia [16], melanoma [17], and glioma [18], and it is considered a potential oncogene [12]. Therefore, based on its ability to alter multiple pathological signals, Gab2 may be an effective target for the treatment of liver diseases.

MicroRNAs are small non-coding RNAs (about 18–25 nucleotides in length) that interact with complementary sequences in the 3′-untranslated regions (UTRs) of target mRNA and inhibit or decrease mRNA translation [19,20]. They are considered to be an important epigenetic regulatory mechanism in eukaryotes. DNA methylation and histone modifications also affect the expression of miRNAs; thus, miRNAs and epigenetics compose an interactive network that regulates the pathogenesis of cancer and many other diseases [21,22]. Current studies have found that miRNAs have critical regulatory effects on the pathogenesis of liver diseases and are involved in metabolism, injury, fibrosis, and tumors in liver tissue. miRNAs not only serve as biomarkers for diagnosis, but also become effective therapeutic targets [19,21,22]. A variety of miRNAs, including miR-122, miR-21, miR-9, miR-181a, and miR-194, are related to insulin resistance and other pathogenic conditions inducing the development of liver diseases [19,21,22]. Many miRNAs are involved in the molecular mechanism of HCC and affect many cancer-related genes, such as *P27*, *P57*, *DIT4*, and *BMF* [19,21,22]. Multiple miRNAs suppress Gab2 transcription in many types of cancer, such as breast cancer, colorectal cancer, glioma cells, and renal cell cancer cells, thus inhibiting the growth and metastasis of cancer cells [23,24,25,26]. miRNA-663 has recently been found to target Gab2 to inhibit cell proliferation and invasion in liver cancer [27]. Therefore, based on our research, we screened for and discovered two new Gab2-targeting miRNAs—miR-181a and miR-9—which are anticipated to improve the prevention and treatment of liver disease as potential biomarkers and therapeutic targets.

## 2. Materials and Methods

### 2.1. Cell Culture and Western Blotting

The normal liver cell line HL-7702 and the hepatocellular carcinoma cell lines HepG2 and SMMC-7721 were obtained from the American Type Culture Collection (ATCC, Rockville, MD, USA) and cultured in RPMI 1640 medium or DMEM (high-glucose) supplemented with 10% FBS, respectively. To evaluate the protein level of Gab2 in liver cells, cells were lysed and the intracellular protein level was checked by Western blotting with an anti-Gab2 antibody (Cell Signaling, 3239, Danvers, MA, USA) according to the description in a previous article [9].

### 2.2. miRNA Level Measurement by Real-Time PCR

Total RNA was extracted from cells with TRIzol reagent (Roche, Basel, Switzerland), and 1 µg of RNA was reverse transcribed into cDNA. Gab2 mRNA and miRNA levels were measured via real-time PCR using an ABI 7500 Sequence Detector System (SYBR Green, Roche, Basel, Switzerland). The primers used for PCR are listed in Appendix A.

### 2.3. Cell Transfection and miRNA Mimics

The mimics and inhibitors of miR-9, miR-34a, miR-181a, and miR-181c were purchased from RiboBio (Guangzhou, China). To check the function of the corresponding miRNAs, these mimics and inhibitors were transfected into HepG2 cells at approximately 80% confluence using Lipofectamine 2000 Transfection Reagent (Thermo, Waltham, MA, USA). Information on the mimics and inhibitors is listed in Appendix A.

### 2.4. miRNA-Gab2 3′UTR Binding Reporter–Luciferase Constructs and Dual-Luciferase Reporter Assay

The 3′UTR sequence of the human Gab2 protein mRNA was found in NCBI, and the corresponding binding site in the target miRNA was defined with a miRNA target gene prediction website. Then, the target fragments of the Gab2 3′UTR containing the binding site of the miRNA were amplified by PCR and were cloned into the luciferase reporter plasmid pmiRGLO to construct the recombinant miRNA-3′UTR binding reporter plasmid. The miRNA- mutated 3′UTR binding reporter plasmids were constructed via overlap extension PCR. The binding site of miR-181a (TGAATGTT) was mutated to CACCGAGG, and the miR-9 binding site was mutated from GTCAAGGACCAGCAAACCAAAGT to GGAGCGGATCCTCACGTTGCCTT. The primers for these constructs are listed in Appendix A.

Transcriptional activity was assayed with a Dual-Luciferase Reporter System (Sigma-Aldrich, Shanghai, China). The reporter constructs and Renilla luciferase vector were transfected into HepG2 cells. After 36 h, the cells were lysed, and the intracellular luciferase activity was measured by an illuminometer (Berthold, Wildbad, Germany).

### 2.5. Cell Growth Assay

Cell growth was assayed via a CCK-8 assay following the description in a previous paper [9]. In brief, approximately 3000 cells were seeded in a 96-well plate and cultured for different times. At each time point, 20 µL of CCK-8 solution was added into the growth medium, and the cells were incubated for 30 min in the dark. Then, visible light absorbance was measured at a wavelength of 450 nm.

### 2.6. Cell Migration Assay

The migration of HepG2 cells was analyzed using Transwell chambers according to a previous description [28]. Frist, 1 × 10^5^ HepG2 cells were cultured in the upper chambers of Transwells in a growth medium at 37 °C with 5% CO_2_ for 28 h. Then, the chambers were washed two times with PBS, and the cells were fixed with formaldehyde for 10 min. After drying, the cells on the upper surface of the chamber membrane were carefully removed using a cotton swab. Then, the chamber was stained with crystal violet for 20 min, washed three times with PBS, and air-dried. The cells on the bottom surface of the chamber membrane were photographed; to quantitatively analyze cell migration, crystal violet in the polycarbonate membrane of the chamber was dissolved in 33% glacial acetic acid, and the light absorbance was measured at a wavelength of 492 nm.

### 2.7. Statistical Analysis

Quantitative experiments were repeated more than three times, and the data are expressed as the means ± SDs. The significance of differences was determined via a two-tailed Student’s *t*-test or a one-way ANOVA using Graphpad Prism (Graphpad software 6.01, San Diego, CA, USA).

## 3. Results

### 3.1. Screening and Identification of Gab2-Related miRNAs

To screen for Gab2-related miRNAs, five websites—miRDB, TargetScan, PicTar, Human miRNA Target, and MicroRNA—were selected. These websites and software were used for reverse prediction miRNAs that might target human Gab2. These miRNAs are listed in Appendix A. No miRNAs were predicted by all five sites (Appendix A). Thus, we selected the miRNAs that were predicted by multiple websites. In addition, our previous study showed that Gab2 is expressed at a low level in normal liver tissue but expressed at abnormally high levels in alcoholic and non-alcoholic fatty liver and liver cancer, proving that Gab2 is a pathological factor related to liver disease [9,10]. Therefore, abnormal miRNA expression in liver cancer and fatty liver reported in the literature was screened (Appendix A) [29]. Based on the above analysis, we preliminarily screened out nine miRNAs most likely to be related to Gab2 expression: miR-9, miR-18a, miR-18b, miR-34a, miR-181a, miR-181c, miR-134, miR-218, and miR-133.

To verify the correlations more clearly between the expression of these miRNAs and that of Gab2, we checked their expression in the HCC cell line HepG2 and the normal hepatic epithelial cell line HL-7702 and confirmed that the Gab2 mRNA expression in HepG2 cells was significantly higher than that in HL-7702 cells (Figure 1). Although miR-218 and miR-18a had expression patterns similar to that of Gab2 (Figure 1), the expression of miR-9, miR-181a, miR-181c, miR-34a, and miR-134 was opposite to that of Gab2, with low expression in HepG2 cells and high expression in HL-7702 cells (Figure 1). In addition, we also checked the expression of Gab2 and selected miRNAs in another hepatocellular carcinoma cell line, SMMC-7721, and confirmed the opposite pattern of expression between Gab2 and these miRNAs (miR-9, miR-181a, and miR-34a) (Appendix A). Generally, miRNA inhibits the expression of a target gene by binding to the mRNA of the target gene through the principle of complete or incomplete complementary base pairing [30]. Therefore, we focused on those miRNAs whose expression was negatively correlated with that of Gab2. These miRNAs would benefit from improving the prevention and treatment of liver diseases by inhibiting the expression of Gab2.

### 3.2. Gab2 Expression in the Hepatoma Cell Line HepG2 Was Inhibited by the miRNA Mimic

To confirm the inhibitory effects of miRNAs on Gab2 protein expression, we transfected the miRNA mimics of miR-9, miR-34a, miR-181a, and miR-181c into HepG2 cells. miRNA mimics are double-stranded miRNA-like RNA fragments, whose 5′-end binds to the complementary sequence in the 3′UTR of the target gene [30]. miRNA mimics are widely used to simulate miRNAs and study the function of miRNAs [31,32]. The levels of the miR-9, miR-34a, miR-181a, and miR-181c mimics were checked by RT-PCR at 24 h after the miRNA mimics were transfected into HepG2 cells, and they were found to be more than one thousand-fold higher than those in empty (nontransfected) cells (Figure 2A). Then, Gab2 expression was assessed in HepG2 cells transfected with the four miRNA mimics for 36 h. The results showed that the Gab2 protein levels in cells transfected with the miR-9 and miR-181a mimics were significantly lower than those in mimic control cells, indicating that miR-9 and miR-181a inhibited Gab2 protein expression (Figure 2B). However, miR-34a and miR-181c had no effect on the protein expression of Gab2 (Figure 2B). These results prove that Gab2 is the downstream target gene of miR-9 and miR-181a. Interestingly, the level of Gab2 mRNA was significantly reduced by the miR-9 mimic (Figure 2D) but not changed by the miR-181a mimic (Figure 2C), suggesting that the regulatory mechanisms by which miR-9 and miR-181a regulate Gab2 expression may be different.

### 3.3. miR-9 and miR-181a Bind to the 3′UTR Region of Gab2

Through the bioinformatics prediction, it was found that these nine miRNAs all contain partially complementary binding sites with the Gab2 3′UTR. To verify this prediction, a dual-luciferase reporter gene system was used. First, the Gab2 3′UTR fragment containing the miRNA binding site was cloned into the luciferase reporter plasmid to construct the recombinant miRNA-Gab2 3′UTR binding reporter plasmid (Figure 3A). When a miRNA binds to this sequence, the expression of the luciferase protein is affected. To test the specificity of this binding, we also constructed a mutation control reporter plasmid with a nonsense mutation in the miRNA-3′UTR binding sites (Figure 3B). Then, the reporter plasmids were transfected into HepG2 cells along with the miR-9 or miR-181a mimic. After 48 h, the luciferase reporter activity was found to be significantly decreased by cotransfection with either the miR-9 or miR-181a mimics (*p* < 0.001) (Figure 3C). The luminescence intensities of cells transfected with miR-9 and miR-181a mimics were only 82.3 and 85.7%, respectively, of those in cells transfected with the control miRNA mimic (Figure 3B). However, when the binding sites were mutated, the luminescence intensity corresponding to luciferase activity was restored (*p* < 0.01) (Figure 3D). Therefore, these results demonstrate that miR-9 and miR-181a interact with the 3′UTR of Gab2 to regulate its expression.

### 3.4. miR-9 and miR-181a Inhibit the Migration and Proliferation of HepG2 Cells

Next, we assessed the effects of miRNAs on the biological function of Gab2 in hepatocellular carcinoma cells. First, the regulatory effect of miR-9 and miR-181a on the migration ability of HepG2 cells was investigated by a Transwell assay. We transfected the miR-9 mimic and miR-181a mimic individually into HepG2 cells. Twenty-four hours later, the cells were resuspended and inoculated in the upper chamber of a Transwell insert (100,000 cells per well). After 28 h of culture, cells in the bottom chamber were fixed and stained. Compared with that in the control group, the numbers of migrated cells transfected with the miR-9 mimic or miR-181a mimic were significantly reduced (Figure 4A). Quantitative analysis showed that the numbers of miR-9 mimic and miR-181a mimic transfected cells that migrated to the bottom chamber were only approximately 37% of the number of cells transfected with the control miRNA mimic (Figure 4B). These results suggest that miR-9 and miR-181a can suppress the migration of HepG2 cells.

Secondly, we investigated the effects of miR-9 and miR-181a on the cell growth of HepG2 cells via a CCK-8 assay. HepG2 cells (around 10,000 per well) were inoculated into a 96-well plate after transfection with the miR-9 mimic or miR-181a mimic. Then, the cell numbers after 0, 1, 2, and 3 days of growth were determined by a CCK-8 assay. At each growth time point, CCK-8 solution was added and incubated for an additional 0.5 h. Then, the absorbance value of the CCK-8 solution in each well was calculated to determine the cell number. The results showed that the growth rate of cells transfected with the miR-9 mimic or miR-181a mimic was significantly lower than that of cells transfected with the control miRNA mimic (Figure 4C,D).

Together, these results suggest that miR-9 and miR-181a inhibit the migration and proliferation of HepG2 cells.

## 4. Discussion

The etiology of liver cancer is very complex and often stems from multiple chronic diseases [1,3]. In European and American patients with liver cancer, glucose and lipid metabolism disorders, alcohol metabolism disorders, hepatitis C virus infection, and HBV infection account for 36.6, 23.5, 22.4, and 6.3% of cases, respectively. Gene mutations account for 3.2% of cases [33]. Therefore, the prevention and treatment of these chronic diseases will effectively reduce the occurrence of liver cancer [1,3,4]. Moreover, the mechanisms with liver lesions induced by various etiologies are different, and a variety of signals are often pooled together to promote the development of the disease [3,5]. Therefore, drugs targeting a single signaling pathway are usually inadequate to prevent and reverse the disease. Our previous study identified a signal integration protein, Gab2, that is specifically and highly expressed in fatty liver, hepatitis, liver cancer, and other pathologies and integrates a variety of signaling pathways to promote the development of the disease [10,11]. However, deletion of the *Gab2* gene or inhibition of the expression of the Gab2 protein expression ameliorates and prevents the pathology of liver disease pathology [8,9,10,11]. Therefore, Gab2 may be a very promising potential target for the prevention and treatment of liver cancer [10]. Here, two miRNAs—miR-9 and miR-181a—were found to target Gab2 that inhibit the migration and proliferation of HepG2 cells, providing a novel potential method to prevent and treat liver diseases, including chronic diseases and cancers of the liver.

miRNAs have been used as new targets for disease treatment and have been studied and applied in various diseases [21,30]. miRNAs are used in early diagnosis, prognosis assessment, and targeted therapy in liver cancer [19,21]. Further studies of miRNAs in the liver will promote the understanding of the pathogenesis of liver diseases and may improve the identification of biomarkers and the development of therapeutic drugs for liver diseases [21,34]. Therefore, the study of miRNAs targeting Gab2 will be more effective in the identification of drug targets for the prevention and therapy of liver cancer. MicroRNAs targeting Gab2 have been found in many tumors and inhibit tumor development. For example, miR-485 targeting of Gab2 inhibits tumor development in colorectal cancer [24]. miR-302a targeting of Gab2 inhibits the carcinogenesis of glioma cells [23]. miR-218b targeting of Gab2 mediates the tumorigenesis in prostate cancer through the PI3K/AKT/GSK-3β pathway [6]. In breast cancer, miR-98-5p and miR-125A-5p target Gab2 to suppress cell growth and invasion but induce cell apoptosis [25,35]. miRNAs that inhibit Gab2 have also been found in ovarian cancer [20] and human renal cell carcinoma [26]. In hepatocellular carcinoma, miR-663b was found to target Gab2 to inhibit cell proliferation and invasion [27]. Here, we identified two novel miRNAs, miR-9 and miR-181a, that target Gab2 to suppress the proliferation and migration of HepG2 cells.

The miR-181 family includes miR181a, miR-181b, miR-181c, and miR-181d, which have conserved structures and similar functions [34]. Mature miR-9 is derived from three independent precursors—miR-9-1, miR-9-2, and miR-9-3 [36]. Aberrant miR-181a and miR-9 expression has been found in various cancers [36]. The expression levels of miR-181a and miR-9 vary in different types of cancer and at different stages of cancer development, leading to their use as potential diagnostic and prognostic markers for cancer [34,36,37]. miR-181a and miR-9 are involved in the whole process of cancer. However, their biological regulatory effects on tumorigenesis are directly related to their different target genes [36,37]. Therefore, more accurately revealing the expression profiles and regulatory functions of miRNAs in carcinogenesis will help us to better identify biomarkers and therapeutic targets for cancer in the future. In liver cancer, miR-181a has been found to play a central role in the malignant transformation of tumors. miR-181a promotes the development of hepatocellular carcinoma by regulating PTEN, CBX7, WIF1, and HOXB5 [38,39,40,41]. However, other reports have indicated that miR-181a is downregulated in HCC and plays a tumor suppressor role targeting cMet, Egr1, and TGF-β [42,43,44]. miR-181a is thought to be a biomarker for the diagnosis and prognosis of hepatocellular carcinoma [45,46,47,48]. miR-181a is deregulated in viral infection, hepatitis, and cirrhosis in patients with hepatocellular carcinoma [49,50], and it is involved in the sorafenib resistance of hepatocellular carcinoma cells [51,52,53]. The findings related to the expression and regulation of miRNA-9 in hepatocellular carcinoma are inconsistent in current reports. miR-9-5p enhances the tumorigenic ability of hepatocellular carcinoma cells by targeting CPEB3 and ESR1 [54,55], but miR-9 plays a tumor suppressor role in hepatocellular carcinoma by regulating IGF2BP1, Sox11, and P21 [56,57,58,59]. miRNA-9 has been identified as a potential biomarker of hepatocellular carcinoma [60,61,62,63,64]. Our study demonstrated that miR-181a and miR-9 target Gab2 and inhibit the proliferation and migration of HCC cells, not only identifying a new protein whose expression is targeted by miR-181a and miR-9, but also providing a new approach to inhibit Gab2 protein expression. Because the expression and function of Gab2 and miRNAs in different hepatocellular carcinoma cell lines are inconsistent, more studies in different HCC cell lines would perfect the understanding of the relationships between Gab2 and miR-181a and miR-9.

## 5. Conclusions

We screened for and identified two miRNAs targeting Gab2; these miRNAs inhibit the expression of the *Gab2* gene and reduce cell proliferation and migration, providing new drug targets for the prevention and treatment of liver cancer.

## Figures and Tables

**Figure 1 genes-13-02152-f001:**
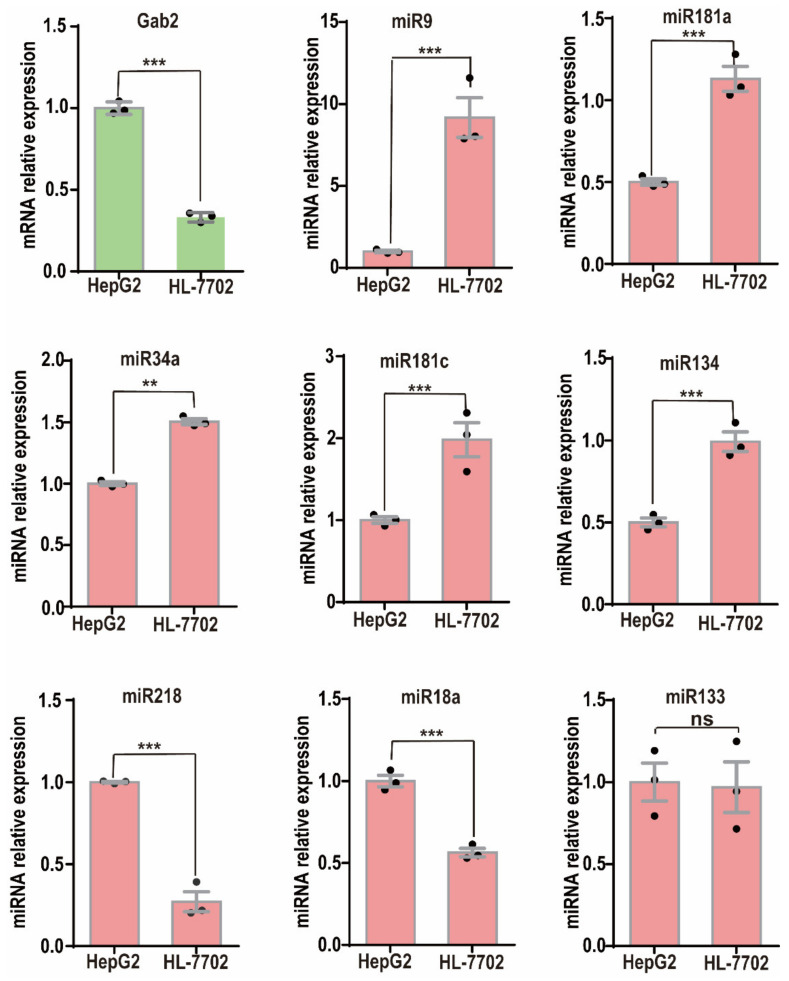
Expression of miRNAs in the hepatocellular carcinoma cell lines HepG2 and HL-7702. The results are presented as the means ± SDs. Statistical comparisons of values were made using Student’s *t*-test. ns, nonsignificant; ** *p* < 0.01, *** *p* < 0.001.

**Figure 2 genes-13-02152-f002:**
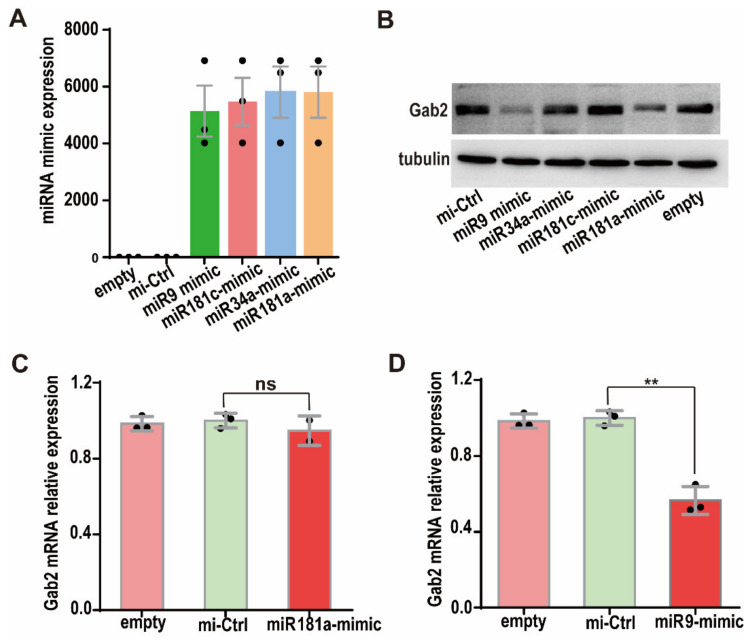
Effect of miRNA mimics transfection on Gab2 protein expression. (**A**) Expression of miRNA mimics. (**B**) Effects of four miRNA mimics on Gab2 protein expression, as checked by Western blotting. Tubulin was used as a total protein control. (**C**,**D**) Effects of hsa-miR-9 mimic and hsa-miR-181a mimic on Gab2, mRNA expression, as analyzed by RT-PCR. Empty: nontransfected; mi-Ctrl: mimic negative control. The results are presented as the means ± SDs. Statistical comparisons of values were made using Student’s *t*-test. ns, nonsignificant; ** *p* < 0.01.

**Figure 3 genes-13-02152-f003:**
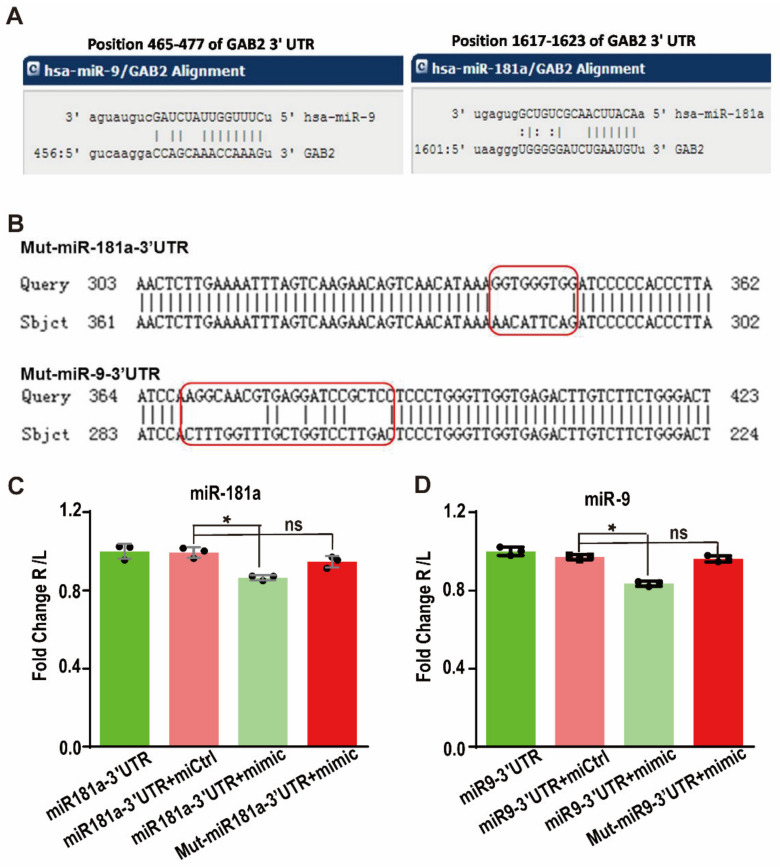
Targeted regulation of Gab2 by hsa-miR-9 and hsa-miR-181a. (**A**) Gab2 binding sites in hsa-miR-9 and hsa-miR-181a predicted by bioinformatics websites. (**B**) miRNAs with mutation of the Gab2 3′UTR binding site. (**C**,**D**) Analysis of the interaction between Gab2 and hsa-miR-9 (**C**) or hsa-miR-181a (**D**) by a dual-luciferase assay. The results are presented as the means ± SDs. Statistical comparisons of values were made using Student’s *t*-test. Red boxs indicate the mutated sequences. ns, nonsignificant; * *p* < 0.05.

**Figure 4 genes-13-02152-f004:**
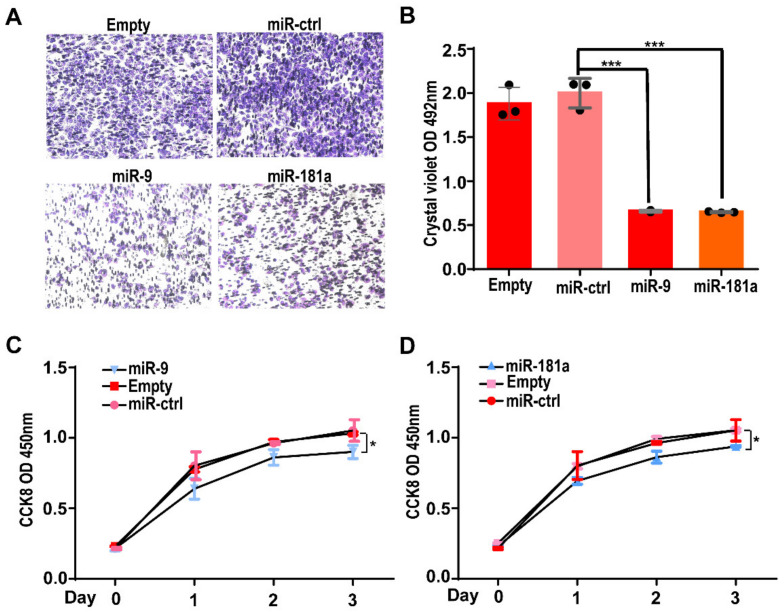
Effects of two miRNAs on the migration and proliferation of HepG2 cells. (**A**) Representative image of migrated HepG2 cells transfected with the hsa-miR-9 or hsa-miR-181a mimic in a Transwell assay. (**B**) The number of migrated cells was determined by measuring the OD value after crystal violet staining. The results are presented as the means ± SDs. Statistical comparisons of values were made using Student’s *t*-test. *** *p* < 0.001. (**C**,**D**) The growth curve of HepG2 cells, as assessed by a CCK-8 assay after transfection with miRNA mimic of the hsa-miR-9 (**C**) or hsa-miR-181a mimic (**D**) for 96 h. The results are presented as the means ± SDs. Statistical comparisons of values were made via one-way ANOVA. * *p* < 0.05.

## Data Availability

All data are contained within the manuscript.

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
