# Peer review of "miR-9 and miR-181a Target Gab2 to Inhibit the Proliferation and Migration of Hepatocellular Carcinoma HepG2 Cells"

_genes, 2022, doi:10.3390/genes13112152_

Round 1
Reviewer 1 Report
In this manuscript, the authors examined the relationship between miRNAs and Gab2. The results showed that miR-9 and miR-181a suppress Gab2 and inhibit the progression of HCC HCC progression by suppressing Gab2.
The strength of the work is the research design, descriptions of methods and results. Overall, I enjoyed reading this article.
First of all, I think the experimental design is appropriate in this study. Also, the introduction and discussion seem to have been cited without excess. One thing that would have made the manuscript more interesting would have been if you had included the changes in the expression levels of Gab2 and each miRNA in different differentiated hepatocellular carcinoma cell lines
There are no significant change points or concerns at this time.
Author Response
Thank you very much for your affirmation of our work and valuable comment. The following is my answer:
Comment 1: English language and style are fine/minor spell check required.
Answer 1: We have polished the revised manuscript by the MDPI English language editing service (English edited 49928).
Comment 2: One thing that would have made the manuscript more interesting would have been if you had included the changes in the expression levels of Gab2 and each miRNA in different differentiated hepatocellular carcinoma cell lines.
Answer 2: Thank you for your kind advice. The expression and function of Gab2 and miRNA in different hepatocellular carcinoma cell lines is inconsistent. Therefore, the investigation of the expression levels of Gab2 and each miRNA in different HCC cell lines would improve and strong the understanding of the relation between Gab2 and miR-181a and miR-9. I have discussed that in the revised manuscription (last four lines in page 9). I am collecting more HCC cell lines and will checked the expression and function of Gab2 and miRNA in future experiments.
Reviewer 2 Report
Major points
In the manuscript “miR-9 and miR-181a target Gab2 to inhibit the proliferation and migration of hepatocellular carcinoma cells” the authors describe two miRNAs indicating their role in the inhibition of proliferation and migration. However, according to the discussion below, for future resubmission more experiments must be performed.
Other groups demonstrated that miR-9 enhances proliferation and migration by targeting genes in hepatocellular carcinoma [PMID: 33106914, PMID: 29487239, PMID: 31531526, PMID: 28520103]. Mir-9 also appears highly expressed in poor prognosis tumors [PMID: 26770365], although in other works has been described as tumor suppressor [PMID: 26547929], for this reason it is not clear its action and the paper must be revised also focusing the attention on Gab2 gene, evaluating/correlating the genes already described.
In addition, miR-181 promotes tumor growth and liver metastasis in colorectal cancer by targeting the tumor suppressor WIF-1, that is important also in HCC.
The result on growth shows a decrease of about <20% in a not defined time (in x-axis there is no value: days, hours) that can be considered a change, but not so strong. Moreover, considering previous works on these miRNAs, the author must check also if Gab2 overexpression with the miRNAs may restore the growth rate.
Figure 1: a real time PCR with TaqMan assay would be more reliable from my point of view (i.e. MiR-181 expression levels increases in HCC compared to non-cancerous tissues [PMID: 35867177])
Figure 2b: insert (also in supplemental materials) a proof that the mimics (different than miR-9 and miR181) are active on another validated targets (i.e. miR34 vs Bcl2, Numb, …)
Minor point
Some flaws must be revised in the text/figures:
i.e. Figure 4 the names in the figures are not aligned.
In the introduction the citation for Gabd2 and HCC [PMID: 28842424] must be inserted.

Author Response
Thank you very much for your constructive and thoughtful comments. The following is my answer:
Comment 1: Moderate English changes required
Answer 1: We have polished the revised manuscript by the MDPI English language editing service (English edited 49928).
Comment 2: Other groups demonstrated that miR-9 enhances proliferation and migration by targeting genes in hepatocellular carcinoma [PMID: 33106914, PMID: 29487239, PMID: 31531526, PMID: 28520103]. Mir-9 also appears highly expressed in poor prognosis tumors [PMID: 26770365], although in other works has been described as tumor suppressor [PMID: 26547929], for this reason it is not clear its action and the paper must be revised also focusing the attention on Gab2 gene, evaluating/correlating the genes already described.
Answer 2: Thank you for your thoughtful suggestion and helpful references. I have cited these references and discussed my results and focused my claim on Gab2 in the revised manuscription (the last paragraph in page 9).
Comment 3: In addition, miR-181 promotes tumor growth and liver metastasis in colorectal cancer by targeting the tumor suppressor WIF-1, that is important also in HCC.
Answer 3: Thank you for your advice. I have cited this reference in revised manuscription (reference #41).
Comment 4: The result on growth shows a decrease of about <20% in a not defined time (in x-axis there is no value: days, hours) that can be considered a change, but not so strong. Moreover, considering previous works on these miRNAs, the author must check also if Gab2 overexpression with the miRNAs may restore the growth rate.
Answer 4: Thank you for your professional advice. I have tried to check if Gab2 overexpression restores the growth rate of cells treated with miRNAs mimic. However, the transfection efficiency of Gab2 in HepG2 is too low to obtain real result. We plan to make a Gab2 stable overexpression cell lines in future experiments.
Comment 5: Figure 1: a real time PCR with TaqMan assay would be more reliable from my point of view (i.e. MiR-181 expression levels increases in HCC compared to non-cancerous tissues [PMID: 35867177])
Answer 5: Thank you for your enlightening advice. RT-PCR with TaqMan assay would be more reliable and improve our results. I will check the mRNA with TaqMan assay in future experiment.
Comment 6: Figure 2b: insert (also in supplemental materials) a proof that the mimics (different than miR-9 and miR181) are active on another validated targets (i.e. miR34 vs Bcl2, Numb, …)
Answer 6: Thank you for your valuable advice. I have cited the proof references in revised manuscription (reference #31,32).
Comment 7: Some flaws must be revised in the text/figures: i.e. Figure 4 the names in the figures are not aligned.
Answer 7: Thank you for your careful work. I have looked through the manuscription and figures and fixed the flaws in Figure 4 and others.
Comment 8: In the introduction the citation for Gabd2 and HCC [PMID: 28842424] must be inserted.
Answer 8: I have cited this reference in revised manuscription (reference #10) following your suggestion.
Reviewer 3 Report
Iantang and colleagues analyzed the expression of several miRNAs in hepatocellular carcinoma. The miRNAs were selected through the use of five different web based tools. Most promissing miRNAs were selected and the effect on Gab2 expression, a protein upregulated in liver carcinoma, was analyzed/inhibited by use of miRNA mimic.
The experiments are well performed the manuscript is clearly structured. A weak point which must be stated in the discussion, is the use of only one HCC cell line. How reliable are the results? Does miR9 and miR181a have same effects also in other liver carcinoma cell lines?
miRNA663 which is mentioned in the introduction part was recently reported to target Gab2. Why miRNA-663 was not analyzed (as a positive control) in the manuscript. Did miRNA-663 did not pop up by using the five web based tools?
Supplemental Figure 1: Insert the numbers of miRNAs in the overlapping fields of the different web tools.
Author Response
Thank you for your constructive and thoughtful comments. The following is my answer:
Comment 1: English language and style are fine/minor spell check required
Answer 1: We have polished the revised manuscript by the MDPI English language editing service (English edited 49928).
Comment 2: he experiments are well performed the manuscript is clearly structured. A weak point which must be stated in the discussion, is the use of only one HCC cell line. How reliable are the results? Does miR9 and miR181a have same effects also in other liver carcinoma cell lines?
Answer 2: Thank you for your instructive suggestion. I have discussed the weak about the use of only one HCC cell line in the revised manuscription (last four lines in page 9) in page 9). The expression and function of Gab2 and miRNA in different hepatocellular carcinoma cell lines is inconsistent. Therefore, the investigation of the expression levels of Gab2 and each miRNA in different HCC cell lines would improve and strong the understanding of the relation between Gab2 and miR-181a and miR-9. I am collecting more HCC cell lines and will checked the expression and function of Gab2 and miRNA in future experiments.
Comment 3: miRNA663 which is mentioned in the introduction part was recently reported to target Gab2. Why miRNA-663 was not analyzed (as a positive control) in the manuscript. Did miRNA-663 did not pop up by using the five web based tools?
Answer 3: Thank you for your nice question. miRNA-663 is a really good positive control for our study. Our experiments started at the early 2018 and I have also read the article in 2019. But, we are sorry we did not employ miRNA-663 as a positive control. miRNA-663 was not predicted by five web based tools.
Comment 4: Supplemental Figure 1: Insert the numbers of miRNAs in the overlapping fields of the different web tools.
Answer 4: I have inserted the numbers of miRNAs in the overlapping fields of the different web tools in revised Supplemental Figure 1.
Round 2
Reviewer 2 Report
the authors plan future experiments to answer, for this reason, the paper in the present form is rejected (EVEN figure 4 was not modified) no additional required experiment was performed, they made just a few changes in the text
Author Response
Comment: the authors plan future experiments to answer, for this reason, the paper in the present form is rejected (EVEN figure 4 was not modified) no additional required experiment was performed, they made just a few changes in the text.
Answer: we are really sorry to disappoint you with our reply. We have been trying to do experiments following the comment of reviewer, but it is a pity that we could not complete them in time. The main author of this manuscript left the laboratory after graduation, and we had to find a new student to train him to do the experiment. New cell lines and some reagents cannot be purchased in time, which is often affected by the COVID-19 epidemic. Now, although the experiment is still not completely complete, we have obtained some data that provide more support to our conclusions:
(1) To solid our finding, we also checked the expression of Gab2 and some miRNA in another hepatocellular carcinoma cell line SMMC-7721 and confirmed the opposite pattern expression between Gab2 and miRNAs (miR-9, miR-181a, miR-34a) (Supplementary Figure S2).
(2) Thank you for your recommendation of TaqMan assay. We tried to check the expression of miRNA with RT-PCR with TaqMan assay. However, we found that there is difficult to observe the signal. The reason may be that the probe could not bind to the miRNA effectively and stably due to too short sequences of miRNA. After many time trying, we obtained the signal at two times (see following figure). But the results are not stable.
(3) We are very sorry that our revisions to the manuscript did not meet your requirements. We have carefully looked through the manuscription again, and fixed the flaws in Figures and texts in the revised manuscript.
